# Natural disasters, livelihood, and healthcare challenges of the people of a riverine island in Bangladesh: A mixed-method exploration

**Ahmed Hossain** [1,2]*, **Anika Tasneem Chowdhury** [3], **Masum Mahbub** [2,4],
**Mahmuda Khan** [2,4], **Taifur Rahman** [3], **Azaz Bin Sharif** [3], **Heba Hijazi** [2,5],
**Mohamad Alameddine** [2]

**1** College of Health Sciences, University of Sharjah, Sharjah, United Arab Emirates, **2** Human Concern USA, Carmel, Indiana, United States of America, **3** Department of Public Health, North South University, Dhaka, Bangladesh, **4** Human Concern International, Ottawa, Canada, **5** Faculty of Medicine, Jordan University of Science and Technology, Irbid, Jordan

* ahmed.hossain@northsouth.edu

**Data Availability Statement:** The complete data, report, inception report, questionnaire and analysis can be found in the source https://osf.io/yj4bt/.

## Abstract

### Background

Bangladesh's islands, because of their geographical location, frequently encounter crises like floods and river erosion, which pose significant threats to the residents' well-being and livelihoods. To delve into the effects of these disasters on livelihood and healthcare challenges, a mixed-method study was undertaken in a riverine-island near a major river of Bangladesh.

### Methodology

Between February 15th and February 28th, 2023, a cross-sectional study was conducted on an island in Bangladesh. The quantitative method involved conducting a survey of 442 households, with a total of 2921 participants. Additionally, 10 in-depth interviews and 10 key-informant interviews were conducted using semi-structured guidelines. Qualitative interviews were audio-recorded, transcribed verbatim, and analyzed using a thematic analysis. Triangulation was employed in this study through the integration of qualitative and quantitative analysis, resulting in the presentation of findings that offer an in-depth comprehension of the phenomenon being investigated.

### Results

River erosions and floods are common and recurring natural disasters that significantly impact the lives of the riverine island inhabitants. These disasters often disrupted their livelihoods, forced many residents to endure substandard living conditions or relocated during flood events. The island faced a low diagnostic prevalence of chronic diseases (e.g., 5.1% of adults were hypertension and 2.5% are diabetes) because of the absence of diagnostic facilities and a shortage of certified doctors. A significant number of chronic illness people in the community turned to alternative medicine sources (39.3%) such as homeopathy,

**Funding:** The author(s) received no specific funding for this work.

**Competing interests:** The authors have declared that no competing interests exist.

Kabiraj, and Ayurvedic medicine, especially it gets increased during periods of natural disasters. Moreover, reproductive aged women revealed that 79.4% of them gave birth at home, with 6.0% of these home deliveries resulting in miscarriage or infant death. The destruction of crops, unstable job opportunities, an inadequate educational system, and a deficient healthcare delivery system exacerbated the hardships faced by the population affected by these disasters.

## Conclusion

The failure to seek treatment for chronic diseases and undiagnosed diseases is a significant health issue among the aging adults on the island. Island residents face the challenge of establishing effective prevention strategies for the well-being of older adults especially at the period of natural disasters. It is crucial for the government and non-governmental organizations (NGOs) to collaborate to prevent the negative effects of floods and river erosions. This should include efforts to enhance the quality of education, healthcare services, job opportunities, and financial assistance for rebuilding homes.

## Introduction

Natural disasters pose a significant threat to global healthcare systems, disrupting access to care, causing shortages of medical supplies and personnel, and exacerbating existing health disparities [1]. These challenges are particularly acute in vulnerable populations, such as those living in low-resource settings or in areas prone to disasters [1,2]. These events can damage healthcare facilities, restrict access to essential services, displace medical workers, and obstruct the delivery of medical care. Furthermore, natural disasters cause scarcities in vital medical supplies, medications, and equipment, leading to a deficiency of trained healthcare workers to provide necessary care [1,3].

Bangladesh is one of the most vulnerable countries to natural disasters due to its geographical location, and socio-economic condition [4]. This is a riverine country that has a complex river system with 800 rivers flowing over its 147,570 square kilometres of surface area [5,6]. Char or riverine island is the by-product of the hydro-morphological dynamics of these rivers which covers about 5% of the total land area of the country and is the habitat of 5% of the total Bangladeshi population [7,8]. These islands are regularly affected by floods, river erosions, and other natural disasters compromising the lives and livelihoods of the dwellers [9]. The main livelihood of this population is agriculture-based and regular damage to their chief source of income forces them to live below the poverty line and sometimes force them to migrate another place [9,10]. The poor communication system keeps these people marginalised from the advantages of mainland Bangladesh creating social, educational, healthcare and financial level inequity [9]. Natural disasters, unemployment, and fiscal deficits have collectively contributed to a significant migration trend among the residents of various islands situated in the North Bengal and Padma River systems of Bangladesh [10]. A study conducted in the Indian state of Jammu and Kashmir revealed that a limited number of partially operational hospitals encountered significant challenges in managing high caseloads and medical equipment destruction following the 2014 heavy flood [11].

The unavailability of healthcare poses a substantial challenge in numerous island and coastal populations inside Bangladesh [12,13]. Remote and isolated regions frequently encounter many challenges pertaining to healthcare, including limited access to medical facilities,

healthcare providers, and essential medicines. These regions commonly experience heat stress and heat-related fatalities, vector-borne diseases like malaria and dengue fever, waterborne diseases, respiratory illnesses, and mental health problems [12–14]. Riverine islands are often home to marginalized and underserved populations. These populations may have less access to education, employment, and other resources, which can further exacerbate health disparities.

While Bangladesh has made significant strides in decreasing maternal and newborn mortality rates [15], there may still be a disparity in mortality rates specifically within the island population compared to the national average. Additionally, the island population might rely more on alternative medications for managing chronic illnesses when compared to the mainland population in Bangladesh. This suggests that healthcare outcomes and practices could differ between island and mainland communities within the country. By studying the healthcare challenges faced by people living on these islands, researchers can gain insights into the challenges faced by people living in other rural or remote areas. This information can then be used to develop interventions to improve the health of these populations.

However, there is an evidence gap regarding healthcare challenges faced by the island population in Bangladesh. Additionally, it remains unclear how natural disasters impact their livelihoods and their ability to access healthcare from the community. This underscores the need for further research and investigation to better understand the unique healthcare dynamics and disaster-related vulnerabilities specific to these island communities.

Health-seeking behavior is a complex and multifaceted phenomenon influenced by various factors. Qualitative methods, such as interviews or focus group discussions, can help explore the depth and context of individuals' experiences and perceptions, while quantitative methods, such as surveys, can provide numerical data on patterns and frequencies of health-seeking behaviors. The combination of both approaches ensures a more comprehensive understanding. The use of both qualitative and quantitative data sources allows for triangulation, where findings from one method can be compared or validated by findings from the other. This enhances the credibility and reliability of the research. Therefore, we conducted a study using both quantitative and qualitative methods of data collection to explore impact of natural disasters on the lives and livelihood of island dwellers and the healthcare challenges faced by the island residents in Bangladesh.

## Methods and materials

### Study design and study site

We conducted a cross-sectional study using a mixed-method approach in Susua Island of Arjuna union of Bhuapur sub-district of Tangail District of Bangladesh from February 15, 2023 to February 28, 2023 where both quantitative and qualitative methods of data collection were used. Tangail district is located between $24^01'$ and $24^047'$ North latitudes and between $89^044'$ and $90^018'$ East longitudes. The Arjuna union is situated at $24^031.28'$ North latitudes and $89^050.53'$ East longitudes. The survey area is Susua island is surrounded by the Jamuna River and the map of the survey area is given in the source https://osf.io/yj4bt/. There are about 1000 households living in the Sosua island. The Susua island is a riverside island that is geographically separated from the mainland. This might pose challenges for individuals residing on the island in accessing necessary healthcare, potentially resulting in various health complications. Through a study of the healthcare obstacles encountered by those residing on this riverine island, researchers can understand the obstacles experienced by individuals living in other rural or isolated regions. Subsequently, these data can be utilized to formulate initiatives aimed at enhancing the well-being of the inhabitants residing in riverine islands.

## Sampling technique and sample size

The sample size for the survey was calculated based on the number of households in Susua village, with a confidence level of 95% and an error margin of 5%. The local community member reported approximately one thousand households in this community. We determined the sample size using the vaccination rate of children in Bangladesh. If we want to be 95% certain that our estimate of the proportion of vaccinated children in the community is within 5% of the true value, and we assumed that 50% of the children in the community are vaccinated, then using finite population correction formula we would need to sample 278 households [16]. The estimated sample size provides the largest sample size necessary for the survey using the proportion of 50%. A sample size of over 400 households is sufficiently large from a homogeneous population, even if the probability of vaccination is smaller or larger than 50%. We finally gathered information from 445 conveniently selected households for the quantitative part of the study.

For the qualitative part of the study, we selected 10 community leaders for key-informant interviews and 10 community people for in-depth interviews. The inclusion and exclusion criteria of the survey participants are:

**Inclusion criteria:**

1. During the study period, at least one adult (18 years) from a household was interviewed.

2. Data on the sociodemographic and health status of the adults was taken if the primary adult (household head) was present in the household during the interview.

3. Data on reproductive health was collected from women only, with no adult male present nearby the respondents.

4. Respondents who gave written informed consent or verbal consent as applicable with the presence of a witness.

**Exclusion criteria:**

1. The participant had just been in residence for less than six months.

2. Participants from households with members over 70 years were not considered in the survey as primary respondents.

3. The households with the existence of patients with high fever or other symptoms of COVID-19 were excluded from the study.

4. Patients who were disabled or receiving palliative care treatment were excluded as primary respondents from the study

## Data collection

Expert data collectors proficient in both quantitative and qualitative research methods utilized structured questionnaires for the quantitative survey and semi-structured interview guidelines for in-depth and key-informant interviews. The interviews covered a range of topics, including socio-demographic information, health status, vaccination status, reproductive health, healthcare service availability, and experiences with natural disasters and their impact on livelihoods.

The survey questionnaire and interview guidelines were initially developed in English, then translated into Bangla by a certified translator. A reverse translation was performed to ensure accuracy, and the final interviews were conducted in Bangla. Pilot testing was carried out on February 13–14, 2023, involving 10 households in Kamrangir Char, Dhaka, with necessary

**Table 1. Distribution of the respondents.**

| Interview type | Male: Female | Role | Total number |
|---|---|---|---|
| Key-Informant Interview (KII) | 7:3 | Madrasa Teacher: 3(M), 1(F); U.P. Member: 1(M), 1(F); Madrasa Principal: 1(M); Religious leader: 1(M), Former Headmistress: 1(F); Primary School Teacher: 1(M) | 10 |
| In-depth Interview (IDI) | 6:4 | Community people: 10 | 10 |
| **Total** | | | **20** |

adjustments made based on feedback. An independent panel of academic and research experts in epidemiology reviewed the questionnaire's validity, assessing its clarity, content, and appropriateness for understanding the community's perspective and cognitive ease. The questionnaire was conducted by an interviewer and the respondent's written consent was acquired. There is a possibility of interviewer bias in this method of data gathering. The collection of information technique involved the recruitment of trained data assistants to mitigate interviewer bias.

The qualitative interviews lasted around 30 minutes. We selected a key informant residing in the community for the past five years and established contact with them via phone before conducting an in-person interview. Key informants were asked to provide a comprehensive overview of the health care, helping to understand the broader context, policies, and systemic factors influencing the study area. In-depth interviews were carried out with residents who had been living in the community for the last five years and were not engaged in community leadership roles. In-depth interviews allowed us to delve deeply into the individual narratives, capturing the nuances and intricacies of participants' experiences for understanding how individuals navigate and make sense of health-related phenomena. All interviews were audio-recorded and transcribed verbatim. The interviews were conducted within the households of the respondents at their convenience, and efforts were made to create an informal and comfortable environment for open and honest responses. The distribution of the respondents can be found in **Table 1**.

## Data analysis

The quantitative survey data was analyzed using R 3.6.3, with categorical factors described through the percentage distribution of participants. Three individuals from 445 households did not respond completely in the quantitative survey. Missing data were excluded from the analysis, and the study utilized available data from 442 households. For the qualitative study, data were complete for all participants. Qualitative data analysis followed a thematic analysis approach, where responses were categorized thematically within and across themes. A single matrix-based coding framework was developed for both the key-informant and in-depth interviews. This codebook was refined through repeated readings of the interview transcripts, leading to the identification of themes and sub-themes. The coding tree is illustrated in Fig 1.

Lastly, the study's findings from both the quantitative and qualitative components were compared and cross-referenced to ensure consistency and validation of the results. The complete data, report, and questionnaire can be found in the source https://osf.io/yj4bt/.

## Results

### Characteristics of the island population

The quantitative survey included 2,921 participants from 442 households, of which 1,872 (62.6%) were adults aged 18 years and above. The average age of the participants was 27.16

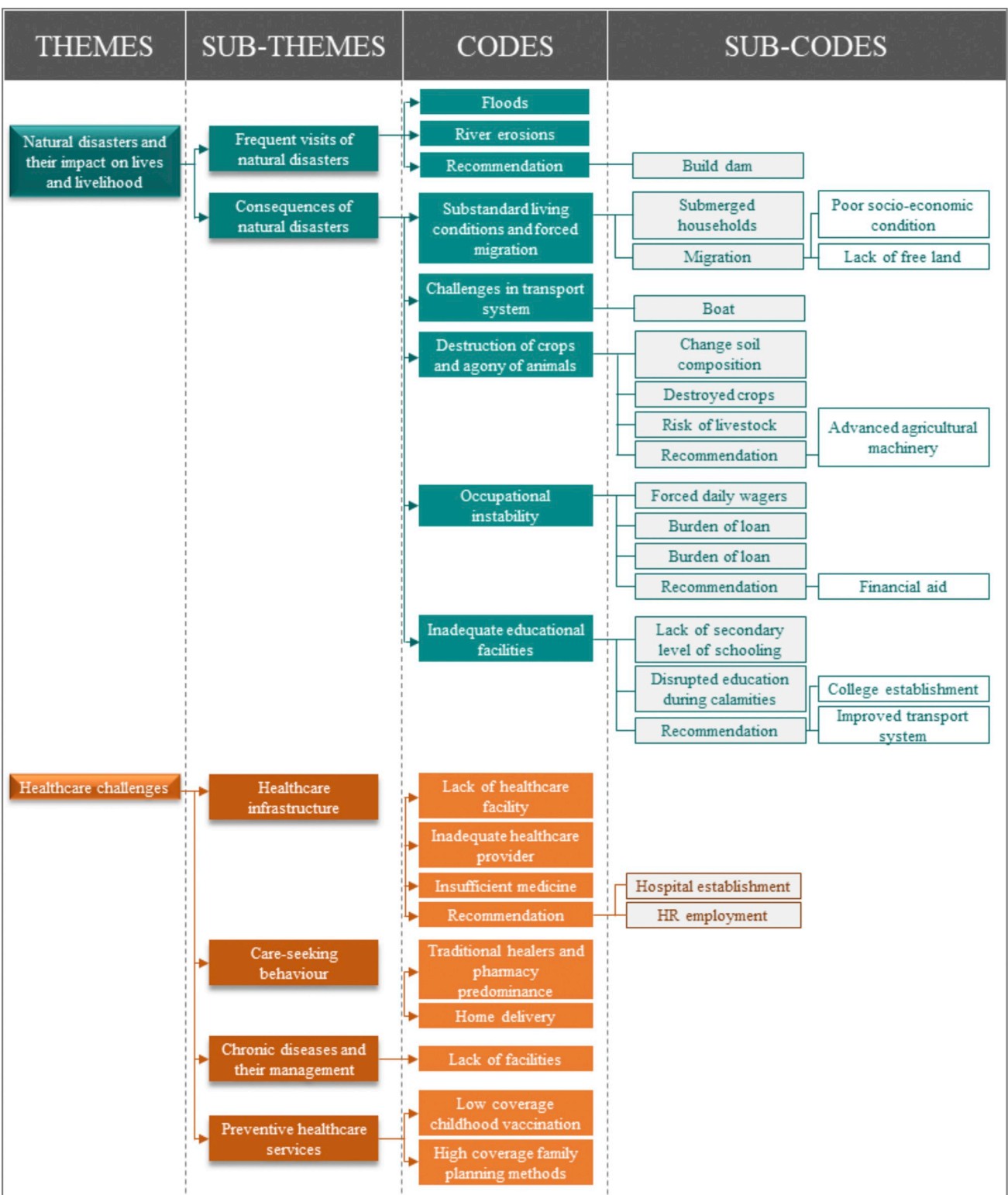

**Fig 1. Coding tree for qualitative analysis.**

**Table 2. Characterisitcs of the participants.**

| Background characteristics | Male | Female | Total |
|---|---|---|---|
| **Age groups** | | | |
| 18–25 years | 277 (28.5%) | 289 (32.1%) | 566 (30.2%) |
| 26–35 years | 239 (24.6%) | 196 (21.8%) | 435 (23.2%) |
| 36–45 years | 171(17.6%) | 188 (20.9%) | 359 (19.2%) |
| 46–55 years | 129 (13.3%) | 118 (13.1%) | 247 (13.2%) |
| 56–65 years | 99 (10.2%) | 73 (8.1%) | 172 (9.2%) |
| 65+ years | 56 (5.8%) | 37 (4.1%) | 93 (5%) |
| **Educational status** | | | |
| No Schooling | 401(41.3%) | 369 (41.3%) | 770 (41.3%) |
| 1–5 Years schooling | 265 (27.3%) | 269 (30.1%) | 534 (28.7%) |
| 6–10 Years schooling | 127 (13.1%) | 170 (19.0%) | 297 (15.9%) |
| More than 10 years schooling | 128 (13.2%) | 48 (5.4%) | 176 (9.4%) |
| Madrasa | 49 (5.1%) | 37 (4.1%) | 86 (4.6%) |
| **Occupational status** | | | |
| Unemployed or not working | 54 (5.6%) | 12 (1.4%) | 66 (3.6%) |
| Housewife | 0 (0%) | 755 (85%) | 755 (40.7%) |
| Student | 73 (7.6%) | 36 (4.1%) | 109 (5.9%) |
| Employed | 838 (86.8%) | 85 (9.6%) | 938 (49.8%) |
| **Marital status** | | | |
| Unmarried | 172 (17.7%) | 45 (5.0%) | 217 (11.6%) |
| Married | 780 (80.4%) | 769 (85.9%) | 1549 (83.1%) |
| Divorced/ Widowed/ Separated | 18 (1.9%) | 81 (9.1%) | 99 (5.3%) |

years, with a standard deviation of 19.23. The characteristics of the participants are given in **Table 2**. Among the participants, 1,482 (50.7%) were male, and 1,439 (49.3%) were female. It was observed that individuals from the community had limited education, with around 41% of them did not attend school or madrasa. A noteworthy demographic point is that one-tenth of the women in the community were divorced or widowed, in contrast to only one percent of the males. Agriculture was the primary occupation in this community, with most adult males working in the field, and many of them also engaged in small businesses. On the other hand, a significant proportion of women, 85%, were housewives, while 87% of adult males were employed in some form of paid work.

## Qualitative survey

A qualitative study was conducted to investigate the effects of natural disasters, livelihood, and healthcare challenges on island-dwelling communities. The research involved 10 key informant interviews with community leaders and 10 in-depth interviews with community members. The participants were selected purposively based on their ability to communicate their feelings effectively and transparently to the interviewer. The thematic analysis from the qualitative survey and triangulation with quantitative analysis are given in the following:

## Natural disasters and their impact on lives and livelihood

**Frequent natural disasters in Char Island.**   The respondents in the study reported experiencing recurrent natural disasters such as severe storms, floods, and river erosion. Specifically, the island they lived on submerged annually between June and August due to flooding. Moreover, river erosion affects their livelihoods every six months, and when combined

with floods, the situation gets even worse. The community members strongly advocated for the construction of a dam to protect them from these frequent disasters. Most of the key informants agreed that the island residents had no option but to relocate far from their homes, often facing challenges related to transportation. A retired head teacher, who had lived on the island for over 40 years, emphasized the dire situation faced by the community:

> "*Our entire farm was submerged in water during last year's flood. It harms the area's property, habitations, and agriculture. Some people have no choice but to seek refuge in school since their homes are uninhabitable. The local road and transit systems are appalling. It might be exceedingly challenging to transport someone who falls ill late at night to Sirajganj or Vuapur (a local Upazila leader and teacher, 51 years).*"

> "*Flooding and river erosion are the primary issues of our Char. Living on this Char terrain during the monsoon is quite challenging. We also transfer our homes from one location to another due to river erosion.*"

(Madrasa Teacher, 38 years)

> "*To prevent the schools from being consumed by the river and to assist the community in avoiding considerable financial loss due to river erosion, a dam must be built.*"

(Community leader, 47 years)

**Consequences of natural disasters.**  *Substandard living conditions and forced migration.* The study revealed that the island's inhabitants faced significant challenges, including substandard living conditions and forced migration. During floods, people were compelled to live in homes partially submerged in water, with water levels reaching their knees or waists. To cope, some raised their beds, others created makeshift hanging bamboo beds called "Macha," and some used rooftops as temporary shelters. However, individuals with limited socio-economic resources were compelled to migrate if their homes were destroyed by floods or river erosion, as there was insufficient available land in their locality for rebuilding.

> "*Flooding and river erosion are the primary issues of our Char. Living on this Char terrain during the monsoon is quite challenging. We also transfer our homes from one location to another due to river erosion.*"

(Madrasa Teacher, 38 years)

*Challenges in the transport system.* During the flood, boats were the primary means of transport. However, the parents were frightened to allow their children to travel alone on the boats.

> "*We have a primary school, but during the flood, our kids cannot get to school without a boat. Children reside in many neighbourhoods or places. It is challenging to transport kids to school from various areas. In these flood times, parents are frightened to leave their kids alone. They worry that the boat will capsize.*"

> "*It is tough to take kids home from day-care. One youngster was left alone in the building. Following that, his parents picked him up from school some hours later.*"

(Community people, Male, 55 years)

A respondent recommended to improve the transport system.

*"Flood is a primary reason why students cannot go to educational institutions. The teachers cannot reach the institution properly in time due to the adverse road system during the flood and rainy season. So, the transportation system could be improved for better access to the education facilities at our Char (island)."*

(Religious leader, Male, 48 years)

*Destruction of crops and agony of animals.* The majority of the residents of the islands relied on the agricultural industry for a living. Heavy storms during the summer season, floods during the rainy season, and six-monthly river erosion damaged agricultural lands. The recent change in the climate had aggravated the problem. According to the island residents, the island had clay soil 20 to 30 years ago making it an ideal land for crop production. However, the replacement of the clay with sandy soil has damaged the land fertility and delayed rain has shortened crop maturation time which had in turn hampered the overall agricultural industry of the island.

*"We plant some crops on our property in the hopes that we can grow them now when there is a lack of rain. Then when heavy rain began to fall all at once, all those crops were destroyed. We are also unable to cultivate rice in our fields due to severe rain that falls at the incorrect time. This scenario is caused by both a lack of rain and heavy rain that fell at the wrong moment."*

(Community leader, 40 years)

Animals also suffer due to floods and river erosion.

*"Animals suffer tremendously during floods, which submerge homes and agricultural land and damage crops."*

(Madrasa superintendent, 28 years)

The respondents recommended to introduce advanced agricultural machinery to support the damaged agricultural system of the island.

*"The Char (Island) people require aid with agriculture. As a result, farmers may cultivate their land with less difficulty. To support their farm, they require advanced agricultural machinery."*

(Primary Head Teacher, 43 years)

*Occupational instability.* The disaster-affected people migrated to work as day labourers at construction sites, brickfields, or as rickshaw pullers in city areas. Some worked as a day labourer on owner's fields for 20 to 30 days and spent the following five months idle. Most households initially survived on loans from banks, NGOs, or other people which later turned into an additional burden to carry on as a 'Year-long installation'.

*"We don't have any work when there's a flood. At that moment, there was nothing we could do. Some individuals rely on the assistance of others, provide money, and get loans from various NGOs. Then they pay this loan in equal instalments over a full year. They obtain a loan for the flood, and the entire year is spent paying off these obligations, preventing them from saving money."*

(Community people, Male, 45 years)

One of the respondents suggested that financial aid and other forms of occupational support would be necessary to help island dwellers combat their occupational instability.

*"Free distribution of seeds for crops and cattle or monetary assistance could help us."*

(Community people, Female, 28 years)

*Inadequate educational facilities.* The island had primary school and madrasas. So, those who completed their primary level of schooling could not continue their education in their locality. Therefore, the community people recommended to establish a high school and college where their children could continue their secondary education. Additionally, when natural disasters stormed the island, the regular educational systems suffered a lot due to the damaged road and poor transport system. A religious leader said:

*"Flood is a primary reason why students cannot go to educational institutions. The teachers cannot reach the institution properly in time due to the adverse road system during the flood and rainy season. So, the transportation system could be improved for better access to the education facilities at our Char."*

## Healthcare challenges

**Healthcare infrastructure in island.** The respondents were unsatisfied with the fragile healthcare service offered on the island. There was only one community clinic that offered primary-level healthcare, care for cough and cold, and fever. This community clinic was located outside the community and the only method of transport was by boat. The healthcare provider was not regular, and the amount of free medicine was insufficient and disproportionate to their population. There was no adequately qualified doctor, and properly equipped hospital/ clinic for the treatment of complicated diseases in the community. Most of the time, the Char people had to go to the city to get healthcare facilities for treatment of severe illness.

The respondents recommended to establish a hospital with qualified doctors to treat emergency and complicated cases.

*"We could benefit much if there was a hospital in the Char with regular visits from qualified doctors. If there were at least two camps per month. It would be a huge assistance to us."*

(Madrasa Teacher, Male, 38 years)

**Chronic diseases among adults in island population.** Fig 2 displays the percentage of adults (18 years and above) in the community with ongoing illnesses diagnosed by a doctor. Despite the absence of healthcare facilities, some critical patients, including those with cancer or who have experienced a stroke, are present in the community. Additionally, residents with common chronic conditions such as hypertension (5%) and diabetes (2%) live in the area, but it remains uncertain whether they have consistent access to the necessary medications. The island faces a low diagnostic prevalence of chronic diseases because of the absence of diagnostic facilities and a shortage of certified doctors. As a result, older adults who are dealing with chronic conditions encounter significant challenges in accessing healthcare services on the island. Furthermore, the relatively low numbers of individuals with hypertension and diabetes in comparison to national levels suggest that many cases of undiagnosed chronic illnesses exist on the island.

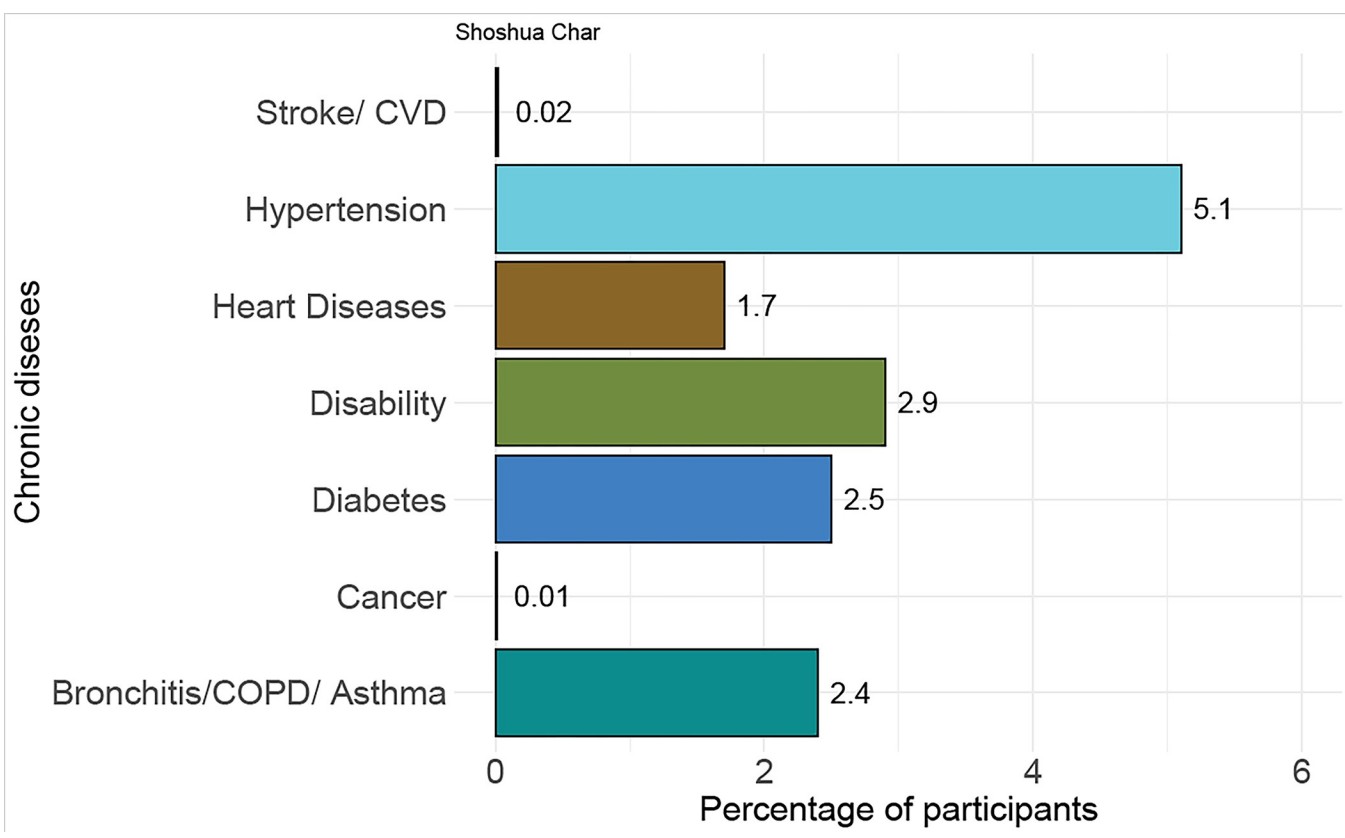

**Fig 2. Prevalence of chronic disease diagnosed by doctor among the island-dwellers.**

Due to the lack of a diagnostic facility and inadequate certified doctors, the diagnostic prevalence of chronic disease is low on the island. Besides, they don't have facilities to treat and manage chronic non-communicable diseases. Older adults suffering from chronic non-communicable diseases face severe difficulties in getting health care. A key informant who is also suffering from hypertension at age 58 mentioned:

*"For such a big population, no hospital or staff is available. We require a free-of-charge hospital in this Char to treat conditions like high blood pressure and diabetes."*

**Care seeking behaviour in island population.** In Fig 3, the care-seeking behaviour of the adults is depicted. The data shows that the majority of the community members (93.3%) rely on pharmacies for disease treatment. About a half of the residents visited either public or private hospitals (57.5%) for their medical needs. Interestingly, a significant number of people in the community turned to alternative medicine sources (39.3%) such as homeopathy, Kabiraj, and Ayurvedic medicine, especially it gets increased during periods of natural disasters. One hypertensive patient at age 58 from the community mentioned,

*"I did not visit a hospital to manage my hypertension but instead purchased medication from a nearby pharmacy. However, during the last flood, I stopped taking my medication to control blood pressure because obtaining medicine became difficult due to the flood conditions."*

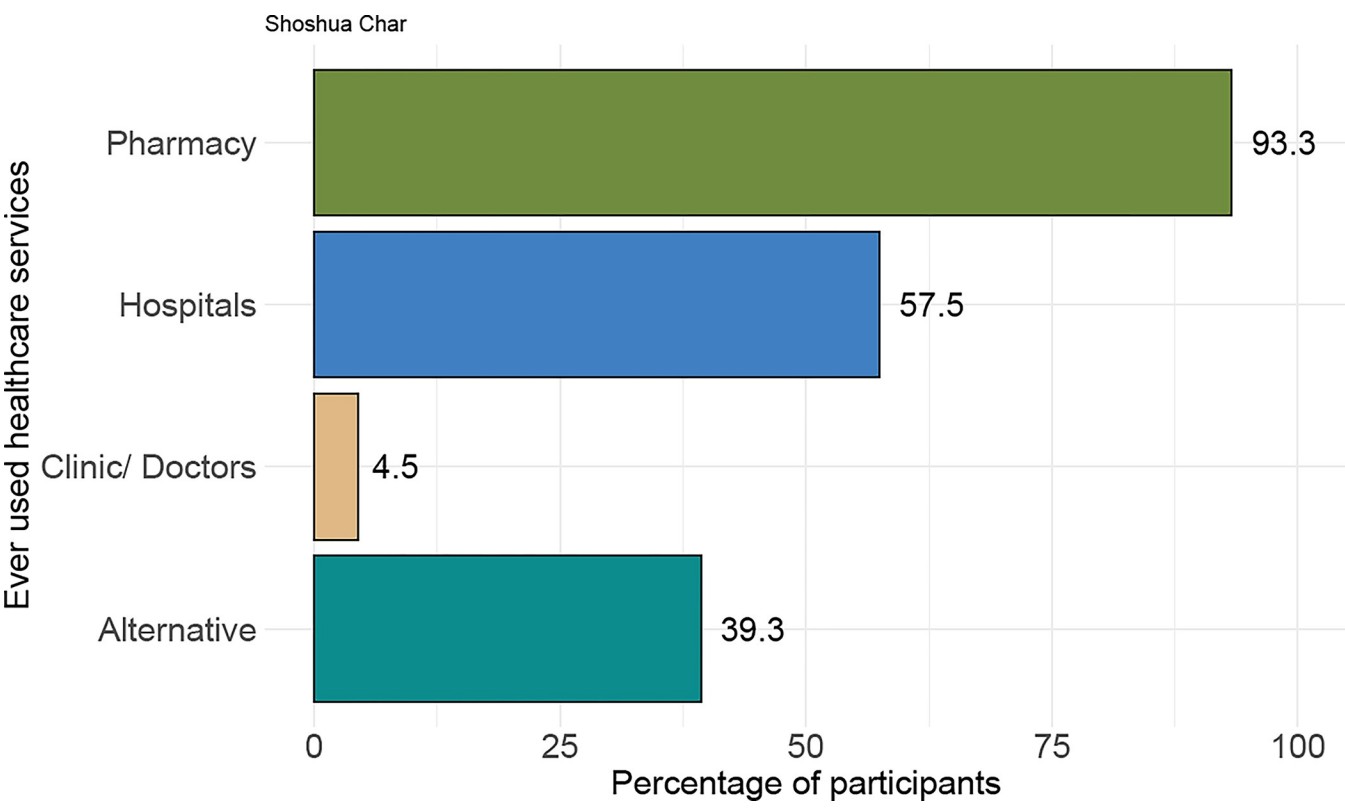

**Fig 3. Healthcare-seeking habits of adult residents who have chronic illnesses.**

**Maternal and child healthcare in island.** A survey of 126 women aged 15–49 years revealed that 79.4% of them gave birth at home, with 6.0% of these home deliveries resulting in miscarriage or infant death. This high rate of home deliveries is concerning due to associated risks like increased maternal and infant mortality and complications. Additionally, out of 154 responding mothers with under-five children, 86.4% reported their children were vaccinated, implying a notable gap between survey and national vaccination rate of about 95%. A woman at age 24 who had a birth at home mentioned,

"*Accessing hospital services for medical needs was challenging due to the need to travel by boat, making it difficult during times of pain or emergencies. Additionally, there is a strong trust within the community in the local birth attendant for healthcare services.*"

## Discussion

This study aimed to investigate the healthcare challenges encountered by the island population in Bangladesh, shedding light on the distinctive healthcare disparities and barriers faced by these communities. We also explore the impact of natural disasters on the lives, livelihoods, and healthcare challenges of an island residents in Bangladesh. The findings revealed that these island dwellers frequently experience natural disasters, including floods and river erosions. Bangladesh is situated within 7.5% of the Ganga, Brahmaputra, and Meghna (GBM) river basins, where heavy monsoon precipitation leads to overbank flooding of riverine islands [17]. A few studies mentioned the climate change with impact of frequent natural disasters and drought in places of Bangladesh [18,19].

Due to the frequent visits of natural disasters, the respondents had to live in a substandard condition or migrate to a different place. Many studies mentioned that migration is a coping mechanism for the environmental and socio-economic stressors created by a natural disaster [18–21].

The respondents in the study reported sufferings from household destruction, crop destruction, and economic instability. The islands are small lands and finding a suitable place to relocate was seldom possible. As a result, the residents had to forcefully migrate to a different area. Those who managed to continue their lives and livelihood in the islands suffered from poor living conditions, lack of employment, and low income. These findings are similar to the finding of studies conducted in islands and coastal areas of Bangladesh [11,12]. A few studies found that inadequate housing, unemployment, food shortage, and land destruction were the major problems faced by the riverine-island dwellers located in the North-Bengal River system [15,22,23]. A study highlighted that health-related adverse outcomes were among the most prevalent adverse occurrences following disasters and another study highlighted mental health symptoms also appeared among individuals after a disaster [24,25].

These islands' geographical isolation and restricted healthcare infrastructure provided obstacles for residents in obtaining comprehensive medical evaluations and diagnoses for chronic health conditions. The national prevalence of hypertension is about 25% [26,27], but the study found a significantly lower hypertension prevalence rate of 5%. This suggests that the study population has a notably lower incidence of high blood pressure compared to the broader national average. Musculoskeletal diseases (MSDs), such as bone injuries or disorders, have been observed to be prevalent among island populations, whereas the mainland population of Bangladesh has recorded an approximate prevalence rate of 35% [28]. However, the investigation of this matter has been impeded due to a lack of adequate facilities. The island population in Bangladesh had a distinct healthcare practices and preferences, including the potential use of alternative medications for managing chronic illnesses. The island population are using more alternative medications to manage chronic illness compared to mainland Bangladesh [29].

In the Susua community, a significant number of individuals heavily rely on pharmacies to address their health issues, with approximately nine out of ten people depending on these facilities. A study cited in the text highlights a similar trend among mainland Bangladeshi people, who frequently visit drug shops without a prescription, often receiving medications without proper counselling [30]. Despite the presence of a community clinic that offers free medicines, the island residences do not seek treatment for chronic disorders there. The absence of trained healthcare providers in the area forces patients to travel to the city for proper medical management. This situation underscores the healthcare challenges faced by the Char community, including limited access to comprehensive healthcare services and a reliance on pharmacies due to these limitations.

Vaccination rates for children under the age of five are notably low in this community. While national vaccination data indicates that over 95% of children in this age group are fully vaccinated [31], this community experiences a lower coverage rate, with more than 10% of children not being fully vaccinated. This situation aligns with lower vaccination rates typically observed in hard-to-reach areas [32]. Several factors may contribute to this gap, including challenges in accessing healthcare, insufficient awareness about vaccination importance, and the influence of religious and cultural beliefs among the island-dwelling population.

Our study found that the percentage of home deliveries was alarmingly high in the community which is congruent to the study findings conducted by a study in the hard-to-reach areas of Bangladesh [33,34]. The high rate of home deliveries is concerning, as home deliveries are associated with a number of risks, including increased risk of complications and in turn

increased risk of maternal and infant deaths [33,35]. The high rate of miscarriages and deaths among home deliveries is also troublesome. Miscarriages and neonatal deaths are often preventable, and the high rate of these outcomes suggests that the lack of adequate healthcare infrastructure, trained medical practitioners, and medications need to be addressed to increase institutional deliveries and the outcome of deliveries.

The study has a limitation in that it was conducted on only one island, making it difficult to generalize the findings to all islands in the system. A more comprehensive study involving multiple islands could improve the applicability of the results. Additionally, the cross-sectional survey might not accurately reflect the long-term situation of the community. However, the study's findings are supported by similar results from other studies conducted on islands in different river systems in the country, which strengthens the credibility of the findings. One of the strengths of the study is that it collected data from both community members and leaders, allowing for a wide range of information related to the study's objectives. Additionally, the use of both qualitative and quantitative data helped reduce bias, enhance validity, and establish the credibility of the study's findings.

## Conclusion

The mixed method approach offers a synergistic combination of qualitative and quantitative methods, providing a more holistic understanding of health-seeking behaviour, investigated the impact of natural disasters on the lives, livelihoods, and healthcare challenges faced by residents of islands in Bangladesh. It found that frequent natural disasters negatively affected various aspects of their lives, including their households, occupations, incomes, education, transport systems, and healthcare delivery. Common problems resulting from these disasters included migration, land and property loss, income reduction, educational disruption, damaged transport infrastructure, and inadequate healthcare services. The study highlighted that the poor healthcare system hindered the management of chronic diseases, emergency treatment, and preventive measures like childhood vaccination. To address these issues, the government and NGOs should collaborate to prevent flood and erosion damage, improve education and healthcare quality, create job opportunities, provide financial aid for housing reconstruction, and promote affordable housing construction. These measures aim to enhance the quality of life for island communities and enhance their resilience to future challenges.

## Author Contributions

**Conceptualization:** Ahmed Hossain, Masum Mahbub, Mahmuda Khan, Taifur Rahman, Mohamad Alameddine.

**Data curation:** Ahmed Hossain.

**Formal analysis:** Ahmed Hossain.

**Investigation:** Ahmed Hossain.

**Methodology:** Ahmed Hossain, Masum Mahbub, Mahmuda Khan, Heba Hijazi, Mohamad Alameddine.

**Project administration:** Ahmed Hossain, Azaz Bin Sharif.

**Resources:** Mahmuda Khan.

**Software:** Ahmed Hossain.

**Supervision:** Ahmed Hossain.

**Validation:** Ahmed Hossain.

**Visualization:** Ahmed Hossain.

**Writing – original draft:** Ahmed Hossain, Anika Tasneem Chowdhury.

**Writing – review & editing:** Ahmed Hossain, Anika Tasneem Chowdhury, Taifur Rahman, Azaz Bin Sharif, Heba Hijazi, Mohamad Alameddine.

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
