## [Decision Letter · Decision Letter 0]

2 Nov 2023

PONE-D-23-30209Natural disasters, livelihood, and healthcare challenges of the people of a riverine island in Bangladesh: a mixed-method explorationPLOS ONE

Dear Dr. Hossain,

Thank you for submitting your manuscript to PLOS ONE. After careful consideration, we feel that it has merit but does not fully meet PLOS ONE’s publication criteria as it currently stands. Therefore, we invite you to submit a revised version of the manuscript that addresses the points raised during the review process.

We look forward to receiving your revised manuscript.

Kind regards,

Md Nazirul Islam Sarker, PhD

Academic Editor

PLOS ONE

4. We note that Appendix in your submission contain [map/satellite] images which may be copyrighted. All PLOS content is published under the Creative Commons Attribution License (CC BY 4.0), which means that the manuscript, images, and Supporting Information files will be freely available online, and any third party is permitted to access, download, copy, distribute, and use these materials in any way, even commercially, with proper attribution. For these reasons, we cannot publish previously copyrighted maps or satellite images created using proprietary data, such as Google software (Google Maps, Street View, and Earth). For more information, see our copyright guidelines: http://journals.plos.org/plosone/s/licenses-and-copyright.

a. You may seek permission from the original copyright holder of Appendix to publish the content specifically under the CC BY 4.0 license.  

Additional Editor Comments:

The author is advised to address all comments of the respective reviewers point-by-point.

Reviewers' comments:

Reviewer's Responses to Questions

**Comments to the Author**

1. Is the manuscript technically sound, and do the data support the conclusions?

Reviewer #1: Yes

Reviewer #2: Partly

Reviewer #3: Yes

2. Has the statistical analysis been performed appropriately and rigorously? 

Reviewer #1: Yes

Reviewer #2: No

Reviewer #3: Yes

3. Have the authors made all data underlying the findings in their manuscript fully available?

Reviewer #1: Yes

Reviewer #2: Yes

Reviewer #3: Yes

4. Is the manuscript presented in an intelligible fashion and written in standard English?

Reviewer #1: Yes

Reviewer #2: Yes

Reviewer #3: Yes

5. Review Comments to the Author

Reviewer #1: This is a good research paper. The idea is nice. Methodology is standard and findings are consistent with the idea and methodology. That's why I have no hesitation in accepting this paper. The research gap is also adequately filled up.

Reviewer #2: No doubt, Bangladesh is one among the countries vulnerable to natural disasters, causing significant losses to the country at micro and macro level. The authors have conducted this study in the said context particularly focused the healthcare challenges. This is a mixed-method study in which the authors have used data integration approach. I will not recommend that the paper should be published in its presents form. Before acceptance the authors are suggestion to revise the manuscript, my comments (both minor and major mixed) are below.

1. I think if the mixed-method word is there, then no need for the word qualitative research in the keywords.

2. Page3: The authors have taken start from the disasters, that I think you went too broad to address the issue.

I think that straightway focus on the healthcare challenges in developing countries, then focus Bangladesh.

The literature that can help: https://www.sciencedirect.com/science/article/pii/S2666623523000077;
https://bangladesh.un.org/en/247856-pre-crisis-assessment-monsoon-flooding-bangladesh-humanitarian-coordination-task-team-hctt

3. About the problem, page 3. The readers will be more interested about the severity of the problem in the study area, I think the authors have failed to do it. You are suggested that statistical facts and figures should be given that how due to the current prevailing problems, the people have been affected and then what are the challenges, then this study in which aspects going to contribute to the literature. The literature that can help not limited to these: https://ddm.portal.gov.bd/sites/default/files/files/ddm.portal.gov.bd/page/b8246646_ee1d_409c_8705_6a670e156936/2023-03-12-07-10-e2326f304f9520077d827656e21cc6f8.pdf

https://www.sciencedirect.com/science/article/pii/S2212420917304065;
https://onlinelibrary.wiley.com/doi/full/10.1111/1477-8947.12250;

https://www.mdpi.com/2071-1050/13/23/13324;
https://www.sciencedirect.com/science/article/pii/S2212420922004290;
https://www.sciencedirect.com/science/article/pii/S2212420923004107.

4. Page 4. The introduction section should also discuss the need for the mixed-methods, and what the methods are already used why the mixed method is needed and how is helpful.

5. What is contribution to the international literature, is there any message for developing countries?

6. Methods section: The study area selection needs more explanation, that why this area/place is selected.

7. Which formula is used in the study for sample size calculation?

8. The participants for qualitative data need more explanation, how they were selected.

9. "The estimated sample size provides the largest sample size necessary for the survey. A sample size of over 400 households is sufficiently large from a homogeneous population, even if the probability of vaccination is smaller or larger than 50%." This statement should be elaborated and justified.

10. Data Collection: Data collection is stating nothing related to collection, it is more instrument validity or removing the biasness. This section should be revisited. I think you can read this paper if suitable to follow in the methods section: https://www.mdpi.com/2227-9032/10/12/2529;
https://pubmed.ncbi.nlm.nih.gov/34607302/

11. Also create a story why the KI, IDI from two groups has been conducted? what are the reasons ? in relationship to the problem of the study.

12. How much data were excluded should be discussed in the methods section.

13. Results: Characteristics of the island population: Better to present these information in Table.

14. Some more analysis is needed in quantitative data analysis part, yes qualitative data if sufficiently addressed.

15. The study is more qualitative, i suggest the authors to focus more on the quantitative data, you have sufficient participants to enrich this segment.

16. The conclusion section may be divided into three sections, that what are main finding, then what are the policy implications and what are the suggestions at the household level, at the end what will be the recommendations for future studies based on the limitations section.

Reviewer #3: The study utilized both quantitative and qualitative methods to collect data, including a survey of 442 households, in-depth interviews, and key-informant interviews. The findings of the study shed light on the significant challenges faced by the riverine island inhabitants due to recurring natural disasters. Overall, I believe the article provides valuable insights into the topic under investigation. After carefully reviewing the manuscript, I believe that substantial improvements are necessary, particularly in the methodology, and discussion sections, in order to enhance the logical flow and overall quality of the study. In light of these observations, I strongly recommend a major revision of the article. With appropriate revisions and improvements in the methodology, and discussion sections, this study has the potential to make a significant contribution to the field.

Page 5: study design

I would encourage authors to justify why Susua Island was selected for this study. For example, you may highlight that this island is located in geographically challenged area, high poverty, low literacy, inadequate medical, transport and other services, etc.

Provide some background of this area. For example, frequent flooding, untimely precipitation, growing season drought, etc.

Sampling techniques and sample size: You may find the household information of your studied community for the Union Parishad. Please check this website (https://bhuapur.tangail.gov.bd) for updated information of demographic and socio-economic status of your study area.

You have provided so many background of sampling techniques. Please revise this sub-section. Highlight how many households your studied, sampling intensity, how you selected. Provide a table of socio-economic characteristics of the studied households.

You have selected 10 community leaders for KII and 10 community people for IDI. Provide a bit information about their profession so that readers can get information about the expertize of the interviewees.

Appendix: Study area map is needed to be improved. Please add latitude and longitude in the map. Indicate your survey location points.

Page 6: Data collection

Please provide the questionnaires to the Appendix. Also include a brief overview of the interview questions in this section.

Page 7: Results

Qualitative survey: Here you have mentioned you investigated effects of natural disasters, livelihood, and healthcare challenges. I think your IDI/KII participants should be from Union and Upazila Disaster Management Committee, Sub-Assistant Agriculture Officer, Union/Upazila Health care officer, Upazila Agriculture and Livelihood Officer, Upazila Livestock Officer, Women elected representative, Union Parishard Chairman, Ward member, Local pharmacy businessman, Teacher, religious leader and the community people.

Please move Table 1 from results to methodology.

Discussion

I think authors discussed the healthcare challenges quite well, while discussion on natural disasters and livelihood needs to improve. Suggested article (DOI:10.9734/ajea/2014/6165 and DOI:10.1016/j.heliyon.2023.e13016) for discussing the livelihood and natural disasters. Drought is also an emerging challenge in Bangladesh. Discuss your qualitative findings focusing the drought impact on livelihoods.

Conclusion

Write two sentences how your study contribute to science and what practical implications of your study has.

6. PLOS authors have the option to publish the peer review history of their article (what does this mean?). If published, this will include your full peer review and any attached files.

Reviewer #1: No

Reviewer #2: No

Reviewer #3: **Yes: **Md Lokman Hossain

---

## [Author Response · Author response to Decision Letter 0]

13 Dec 2023

December 13, 2023

Md Nazirul Islam Sarker, PhD

Academic Editor

PLOS ONE

Ms. No.: PONE-D-23-30209

Title: Natural disasters, livelihood, and healthcare challenges of the people of a riverine island in Bangladesh: a mixed-method exploration

Dear Professor Md Nazirul Islam Sarker,

Thank you very much for your editorial suggestions and the reviewers’ comments. They were accommodating. Please find enclosed an itemized list of responses along with the revised manuscript. 

In our response to the reviewer, we used regular font for the comments/questions by the referees and regular, bold font for our responses, which are shown immediately following the questions/comments.

Thank you once again for the opportunity to submit a revised manuscript.

Ahmed Hossain, PhD

Professor, Health Care Management

Director, Global Health Institute

University of Sharjah, Sharjah, UAE.

 

Reviewer #1: 

This is a good research paper. The idea is nice. Methodology is standard and findings are consistent with the idea and methodology. That's why I have no hesitation in accepting this paper. The research gap is also adequately filled up.

Authors: Thank you very much. 

 

Reviewer #2: 

No doubt, Bangladesh is one among the countries vulnerable to natural disasters, causing significant losses to the country at micro and macro level. The authors have conducted this study in the said context particularly focused the healthcare challenges. This is a mixed-method study in which the authors have used data integration approach. I will not recommend that the paper should be published in its presents form. Before acceptance the authors are suggestion to revise the manuscript, my comments (both minor and major mixed) are below.

1. I think if the mixed-method word is there, then no need for the word qualitative research in the keywords.

Authors: We have removed the word qualitative research from the Keywords.

2. Page3: The authors have taken start from the disasters, that I think you went too broad to address the issue. I think that straightway focus on the healthcare challenges in developing countries, then focus Bangladesh.

The literature that can help: https://www.sciencedirect.com/science/article/pii/S2666623523000077;
https://bangladesh.un.org/en/247856-pre-crisis-assessment-monsoon-flooding-bangladesh-humanitarian-coordination-task-team-hctt

Authors: Thank you very much for your wonderful suggestion. We have changed the first paragraph of the introduction, and the included a new paragraph as follows. 

“Natural disasters pose a significant threat to global healthcare systems, disrupting access to care, causing shortages of medical supplies and personnel, and exacerbating existing health disparities [1]. These challenges are particularly acute in vulnerable populations, such as those living in low-resource settings or in areas prone to disasters [1-2]. These events can damage healthcare facilities, restrict access to essential services, displace medical workers, and obstruct the delivery of medical care. Furthermore, natural disasters cause scarcities in vital medical supplies, medications, and equipment, leading to a deficiency of trained healthcare workers to provide necessary care [1,3].”

3. About the problem, page 3. The readers will be more interested about the severity of the problem in the study area, I think the authors have failed to do it. You are suggested that statistical facts and figures should be given that how due to the current prevailing problems, the people have been affected and then what are the challenges, then this study in which aspects going to contribute to the literature. The literature that can help not limited to these: https://ddm.portal.gov.bd/sites/default/files/files/ddm.portal.gov.bd/page/b8246646_ee1d_409c_8705_6a670e156936/2023-03-12-07-10-e2326f304f9520077d827656e21cc6f8.pdf

https://www.sciencedirect.com/science/article/pii/S2212420917304065;
https://onlinelibrary.wiley.com/doi/full/10.1111/1477-8947.12250;

https://www.mdpi.com/2071-1050/13/23/13324;
https://www.sciencedirect.com/science/article/pii/S2212420922004290;
https://www.sciencedirect.com/science/article/pii/S2212420923004107.

Authors: Thank you very much for your suggestions. We included the following sentences to understand the depth of the problem.

Riverine islands are often isolated and have limited access to healthcare resources. This can make it difficult for people living on these islands to get the healthcare they need and can lead to a number of health problems. Riverine islands are often home to marginalized and underserved populations. These populations may have less access to education, employment, and other resources, which can further exacerbate health disparities.

Riverine islands can be used as a microcosm for larger populations. By studying the healthcare challenges faced by people living on riverine islands, researchers can gain insights into the challenges faced by people living in other rural or remote areas.

By studying the healthcare challenges faced by people living on these islands, researchers can gain insights into the challenges faced by people living in other rural or remote areas. This information can then be used to develop interventions to improve the health of these populations.

4. Page 4. The introduction section should also discuss the need for the mixed-methods, and what the methods are already used why the mixed method is needed and how is helpful.

Authors: The following paragraph is added in the manuscript to understand the need for the mixed method:

“Health-seeking behavior is a complex and multifaceted phenomenon influenced by various factors. Qualitative methods, such as interviews or focus group discussions, can help explore the depth and context of individuals' experiences and perceptions, while quantitative methods, such as surveys, can provide numerical data on patterns and frequencies of health-seeking behaviors. The combination of both approaches ensures a more comprehensive understanding. The use of both qualitative and quantitative data sources allows for triangulation, where findings from one method can be compared or validated by findings from the other. This enhances the credibility and reliability of the research. Therefore, we conducted a study using both quantitative and qualitative methods of data collection to explore impact of natural disasters on the lives and livelihood of island dwellers and the healthcare challenges faced by the island residents in Bangladesh.” 

5. What is contribution to the international literature, is there any message for developing countries?

Authors: Many thanks for your suggestion. We included a number of articles that features healthcare challenges in international arena:

Chatterjee P. Jammu and Kashmir faces daunting health challenges after floods. BMJ. 2014 Sep 22;349:g5792. doi: 10.1136/bmj.g5792. PMID: 25248452.

Quintussi M, Van de Poel E, Panda P, Rutten F. Economic consequences of ill-health for households in northern rural India. BMC Health Serv Res. 2015 Apr 26;15:179. doi: 10.1186/s12913-015-0833-0. PMID: 25928097; PMCID: PMC4419476.

6. Methods section: The study area selection needs more explanation, that why this area/place is selected.

Authors: Thank you. We included the following sentences in the article.

The Susua island is a riverside island that is geographically separated from the mainland. This might pose challenges for individuals residing on the island in accessing necessary healthcare, potentially resulting in various health complications. Through a study of the healthcare obstacles encountered by those residing on this riverine island, researchers can understand the obstacles experienced by individuals living in other rural or isolated regions. Subsequently, these data can be utilized to formulate initiatives aimed at enhancing the well-being of the inhabitants residing in riverine islands.

7. Which formula is used in the study for sample size calculation?

Authors: The finite population correction formula was used to calculate the sample size. It has been mentioned in the article with a reference. 

8. The participants for qualitative data need more explanation, how they were selected.

Authors: The research involved 10 key informant interviews with community leaders and 10 in-depth interviews with community members. The participants were selected purposively based on their ability to communicate their feelings effectively and transparently to the interviewer. It took about thirty minutes for an interview with every patient. The distribution of the respondents can be found in Table 1. 

Table 1: Distribution of the respondents

Interview type Male: Female Role Total number

Key-Informant Interview (KII) 7:3 Madrasa Teacher: 3(M), 1(F); U.P. Member: 1(M), 1(F); Madrasa Principal: 1(M); Religious leader: 1(M), Former Headmistress: 1(F); Primary School Teacher: 1(M) 10

In-depth Interview (IDI) 6:4 Community people: 10 10

Total 20

9. "The estimated sample size provides the largest sample size necessary for the survey. A sample size of over 400 households is sufficiently large from a homogeneous population, even if the probability of vaccination is smaller or larger than 50%." This statement should be elaborated and justified.

Authors: Thank you for your inquiry. The clarification for the statement above is as follows:

We initially calculated the sample size to be 278, considering the finite population correction. Without applying the finite population correction, the estimated sample size would have been 384, using the formula ( ), where the proportion is assumed to be 0.50. It's worth noting that using a proportion of 0.50 consistently results in the highest sample size. If the proportion deviates from 0.50, either being lower or higher, the necessary sample size would be less than 384.

10. Data Collection: Data collection is stating nothing related to collection, it is more instrument validity or removing the biasness. This section should be revisited. I think you can read this paper if suitable to follow in the methods section: https://www.mdpi.com/2227-9032/10/12/2529;
https://pubmed.ncbi.nlm.nih.gov/34607302/

Authors: Thank you. The data collection is now written as following:

Expert data collectors proficient in both quantitative and qualitative research methods utilized structured questionnaires for the quantitative survey and semi-structured interview guidelines for in-depth and key-informant interviews. The interviews covered a range of topics, including socio-demographic information, health status, vaccination status, reproductive health, healthcare service availability, and experiences with natural disasters and their impact on livelihoods.

The survey questionnaire and interview guidelines were initially developed in English, then translated into Bangla by a certified translator. A reverse translation was performed to ensure accuracy, and the final interviews were conducted in Bangla. Pilot testing was carried out on February 13-14, 2023, involving 10 households in Kamrangir Char, Dhaka, with necessary adjustments made based on feedback. An independent panel of academic and research experts in epidemiology reviewed the questionnaire's validity, assessing its clarity, content, and appropriateness for understanding the community's perspective and cognitive ease. The questionnaire was conducted by an interviewer and the respondent's written consent was acquired. There is a possibility of interviewer bias in this method of data gathering. The collection of information technique involved the recruitment of trained data assistants to mitigate interviewer bias.

The qualitative interviews lasted around 30 minutes. We selected a key informant residing in the community for the past five years and established contact with them via phone before conducting an in-person interview. Key informants were asked to provide a comprehensive overview of the health care, helping to understand the broader context, policies, and systemic factors influencing the study area. In-depth interviews were carried out with residents who had been living in the community for the last five years and were not engaged in community leadership roles. In-depth interviews allowed us to delve deeply into the individual narratives, capturing the nuances and intricacies of participants' experiences for understanding how individuals navigate and make sense of health-related phenomena. All interviews were audio-recorded and transcribed verbatim. The interviews were conducted within the households of the respondents at their convenience, and efforts were made to create an informal and comfortable environment for open and honest responses.

11. Also create a story why the KI, IDI from two groups has been conducted? what are the reasons? in relationship to the problem of the study.

Authors: Thank you. Key informants were asked to provide a comprehensive overview of the health care, helping to understand the broader context, policies, and systemic factors influencing the study area. In-depth interviews allowed us to delve deeply into the individual narratives, capturing the nuances and intricacies of participants' experiences for understanding how individuals navigate and make sense of health-related phenomena.

12. How much data were excluded should be discussed in the methods section.

Authors: Three individuals from 445 households did not respond completely in the quantitative survey. Missing data were excluded from the analysis, and the study utilized available data from 442 households. For the qualitative study, data were complete for all participants.

13. Results: Characteristics of the island population: Better to present these information in Table.

Authors: Thank you. A table has been presented in the manuscript.

Background characteristics Male Female Total

Age groups 

18-25 years 277 (28.5%) 289 (32.1%) 566 (30.2%)

26-35 years 239 (24.6%) 196 (21.8%) 435 (23.2%)

36-45 years 171(17.6%) 188 (20.9%) 359 (19.2%)

46-55 years 129 (13.3%) 118 (13.1%) 247 (13.2%)

56-65 years 99 (10.2%) 73 (8.1%) 172 (9.2%)

65+ years 56 (5.8%) 37 (4.1%) 93 (5%)

Educational status 

No Schooling 401(41.3%) 369 (41.3%) 770 (41.3%)

1-5 Years schooling 265 (27.3%) 269 (30.1%) 534 (28.7%)

6-10 Years schooling 127 (13.1%) 170 (19.0%) 297 (15.9%)

More than 10 years schooling 128 (13.2%) 48 (5.4%) 176 (9.4%)

Madrasa 49 (5.1%) 37 (4.1%) 86 (4.6%)

Occupational status 

Unemployed or not working 54 (5.6%) 12 (1.4%) 66 (3.6%)

Housewife 0 (0%) 755 (85%) 755 (40.7%)

Student 73 (7.6%) 36 (4.1%) 109 (5.9%)

Employed 838 (86.8%) 85 (9.6%) 938 (49.8%)

Marital status 

Unmarried 172 (17.7%) 45 (5.0%) 217 (11.6%)

Married 780 (80.4%) 769 (85.9%) 1549 (83.1%)

Divorced/ Widowed/ Separated 18 (1.9%) 81 (9.1%) 99 (5.3%)

14. Some more analysis is needed in quantitative data analysis part, yes qualitative data if sufficiently addressed.

Authors: Thank you for your suggestion. The demographic information of the community is presented in the manuscript from the quantitive part. 

15. The study is more qualitative, i suggest the authors to focus more on the quantitative data, you have sufficient participants to enrich this segment.

Authors: Thanks. We agree the study is more qualitative. For interested users for the quantitative part, we gave an address to find the report. 

16. The conclusion section may be divided into three sections, that what are main finding, then what are the policy implications and what are the suggestions at the household level, at the end what will be the recommendations for future studies based on the limitations section.

Authors: The current conclusion highlighted all the three things that you mentioned. Please find it in the following:

The mixed method approach offers a synergistic combination of qualitative and quantitative methods, providing a more holistic understanding of health-seeking behaviour, investigated the impact of natural disasters on the lives, livelihoods, and healthcare challenges faced by residents of islands in Bangladesh. It found that frequent natural disasters negatively affected various aspects of their lives, including their households, occupations, incomes, education, transport systems, and healthcare delivery. Common problems resulting from these disasters included migration, land and property loss, income reduction, educational disruption, damaged transport infrastructure, and inadequate healthcare services. The study highlighted that the poor healthcare system hindered the management of chronic diseases, emergency treatment, and preventive measures like childhood vaccination. To address these issues, the government and NGOs should collaborate to prevent flood and erosion damage, improve education and healthcare quality, create job opportunities, provide financial aid for housing reconstruction, and promote affordable housing construction. These measures aim to enhance the quality of life for island communities and enhance their resilience to future challenges.

 

Reviewer #3: 

The study utilized both quantitative and qualitative methods to collect data, including a survey of 442 households, in-depth interviews, and key-informant interviews. The findings of the study shed light on the significant challenges faced by the riverine island inhabitants due to recurring natural disasters. Overall, I believe the article provides valuable insights into the topic under investigation. After carefully reviewing the manuscript, I believe that substantial improvements are necessary, particularly in the methodology, and discussion sections, in order to enhance the logical flow and overall quality of the study. In light of these observations, I strongly recommend a major revision of the article. With appropriate revisions and improvements in the methodology, and discussion sections, this study has the potential to make a significant contribution to the field.

Page 5: study design

I would encourage authors to justify why Susua Island was selected for this study. For example, you may highlight that this island is located in geographically challenged area, high poverty, low literacy, inadequate medical, transport and other services, etc.

Provide some background of this area. For example, frequent flooding, untimely precipitation, growing season drought, etc.

Authors: Thank you. We included the following sentences in the article.

The survey area is Susua island is surrounded by the Jamuna River. There are about 1000 households living in the Sosua island. The Susua island is a riverside island that is geographically separated from the mainland. This might pose challenges for individuals residing on the island in accessing necessary healthcare, potentially resulting in various health complications. Through a study of the healthcare obstacles encountered by those residing on this riverine island, researchers can understand the obstacles experienced by individuals living in other rural or isolated regions. Subsequently, these data can be utilized to formulate initiatives aimed at enhancing the well-being of the inhabitants residing in riverine islands. 

Sampling techniques and sample size: You may find the household information of your studied community for the Union Parishad. Please check this website (https://bhuapur.tangail.gov.bd) for updated information of demographic and socio-economic status of your study area.

Authors: Thank you very much for the link. 

You have provided so many background of sampling techniques. Please revise this sub-section. Highlight how many households your studied, sampling intensity, how you selected. Provide a table of socio-economic characteristics of the studied households.

Authors: Many thanks for your comment. The sampling techniques are described in the manuscript in page 5-6. It was mentioned that 445 households were studied. And the socio-economic characteristics of the studied households were added in the manuscript. 

Background characteristics Male Female Total

Age groups 

18-25 years 277 (28.5%) 289 (32.1%) 566 (30.2%)

26-35 years 239 (24.6%) 196 (21.8%) 435 (23.2%)

36-45 years 171(17.6%) 188 (20.9%) 359 (19.2%)

46-55 years 129 (13.3%) 118 (13.1%) 247 (13.2%)

56-65 years 99 (10.2%) 73 (8.1%) 172 (9.2%)

65+ years 56 (5.8%) 37 (4.1%) 93 (5%)

Educational status 

No Schooling 401(41.3%) 369 (41.3%) 770 (41.3%)

1-5 Years schooling 265 (27.3%) 269 (30.1%) 534 (28.7%)

6-10 Years schooling 127 (13.1%) 170 (19.0%) 297 (15.9%)

More than 10 years schooling 128 (13.2%) 48 (5.4%) 176 (9.4%)

Madrasa 49 (5.1%) 37 (4.1%) 86 (4.6%)

Occupational status 

Unemployed or not working 54 (5.6%) 12 (1.4%) 66 (3.6%)

Housewife 0 (0%) 755 (85%) 755 (40.7%)

Student 73 (7.6%) 36 (4.1%) 109 (5.9%)

Employed 838 (86.8%) 85 (9.6%) 938 (49.8%)

Marital status 

Unmarried 172 (17.7%) 45 (5.0%) 217 (11.6%)

Married 780 (80.4%) 769 (85.9%) 1549 (83.1%)

Divorced/ Widowed/ Separated 18 (1.9%) 81 (9.1%) 99 (5.3%)

You have selected 10 community leaders for KII and 10 community people for IDI. Provide a bit information about their profession so that readers can get information about the expertize of the interviewees.

Authors: Thank you for your suggestion. The distribution of the respondents for the qualitative survey was given in the following. 

Table 1: Distribution of the respondents

Interview type Male: Female Role Total number

Key-Informant Interview (KII) 7:3 Madrasa Teacher: 3(M), 1(F); U.P. Member: 1(M), 1(F); Madrasa Principal: 1(M); Religious leader: 1(M), Former Headmistress: 1(F); Primary School Teacher: 1(M) 10

In-depth Interview (IDI) 6:4 Community people: 10 10

Total 20

Appendix: Study area map is needed to be improved. Please add latitude and longitude in the map. Indicate your survey location points.

Authors: It is mentioned in the manuscript that the Tangail district is located between 2401’ and 24047’ North latitudes and between 89044’ and 90018’ East longitudes. The Arjuna union is situated at 24031.28’ North latitudes and 89050.53’ East longitudes. The map is in the source https://osf.io/yj4bt/ and so it is not required to put latitude and longitudes in the map. The map is only provided to understand that the survey area is surrounded by a river which is clearly identifiable. 

Page 6: Data collection

Please provide the questionnaires to the Appendix. Also include a brief overview of the interview questions in this section.

Authors: Thanks. It was mentioned in the manuscript that the complete data, report, and questionnaire can be found in the source https://osf.io/yj4bt/.

Page 7: Results

Qualitative survey: Here you have mentioned you investigated effects of natural disasters, livelihood, and healthcare challenges. I think your IDI/KII participants should be from Union and Upazila Disaster Management Committee, Sub-Assistant Agriculture Officer, Union/Upazila Health care officer, Upazila Agriculture and Livelihood Officer, Upazila Livestock Officer, Women elected representative, Union Parishard Chairman, Ward member, Local pharmacy businessman, Teacher, religious leader and the community people.

Authors: Many thanks for your suggestion. Many of the roles are not present in the village. Therefore we interviewed the community leaders according to the following distribution. 

Interview type Male: Female Role Total number

Key-Informant Interview (KII) 7:3 Madrasa Teacher: 3(M), 1(F); U.P. Member: 1(M), 1(F); Madrasa Principal: 1(M); Religious leader: 1(M), Former Headmistress: 1(F); Primary School Teacher: 1(M) 10

In-depth Interview (IDI) 6:4 Community people: 10 10

Total 20

Please move Table 1 from results to methodology.

Authors: Thanks. Done. 

Discussion

I think authors discussed the healthcare challenges quite well, while discussion on natural disasters and livelihood needs to improve. Suggested article (DOI:10.9734/ajea/2014/6165 and DOI:10.1016/j.heliyon.2023.e13016) for discussing the livelihood and natural disasters. Drought is also an emerging challenge in Bangladesh. Discuss your qualitative findings focusing the drought impact on livelihoods.

Authors: Thank you for the references. Both the articles are referenced in the discussion section. 

Conclusion

Write two sentences how your study contribute to science and what practical implications of your study has.

Authors: Thank you for your comment. Taking consideration of the comment from second reviewer we wrote our conclusion. 

The mixed method approach offers a synergistic combination of qualitative and quantitative methods, providing a more holistic understanding of health-seeking behaviour, investigated the impact of natural disasters on the lives, livelihoods, and healthcare challenges faced by residents of islands in Bangladesh. It found that frequent natural disasters negatively affected various aspects of their lives, including their households, occupations, incomes, education, transport systems, and healthcare delivery. Common problems resulting from these disasters included migration, land and property loss, income reduction, educational disruption, damaged transport infrastructure, and inadequate healthcare services. The study highlighted that the poor healthcare system hindered the management of chronic diseases, emergency treatment, and preventive measures like childhood vaccination. To address these issues, the government and NGOs should collaborate to prevent flood and erosion damage, improve education and healthcare quality, create job opportunities, provide financial aid for housing reconstruction, and promote affordable housing construction. These measures aim to enhance the quality of life for island communities and enhance their resilience to future challenges.

---

## [Decision Letter · Decision Letter 1]

1 Feb 2024

Natural disasters, livelihood, and healthcare challenges of the people of a riverine island in Bangladesh: a mixed-method exploration

PONE-D-23-30209R1

Dear Dr. Hossain,

We’re pleased to inform you that your manuscript has been judged scientifically suitable for publication and will be formally accepted for publication once it meets all outstanding technical requirements.

Kind regards,

Mohammed Sarfaraz Gani Adnan, PhD

Academic Editor

PLOS ONE

Additional Editor Comments (optional):

Reviewers' comments:

Reviewer's Responses to Questions

**Comments to the Author**

1. If the authors have adequately addressed your comments raised in a previous round of review and you feel that this manuscript is now acceptable for publication, you may indicate that here to bypass the “Comments to the Author” section, enter your conflict of interest statement in the “Confidential to Editor” section, and submit your "Accept" recommendation.

Reviewer #1: All comments have been addressed

Reviewer #3: All comments have been addressed

2. Is the manuscript technically sound, and do the data support the conclusions?

Reviewer #1: Yes

Reviewer #3: Yes

3. Has the statistical analysis been performed appropriately and rigorously? 

Reviewer #1: Yes

Reviewer #3: N/A

4. Have the authors made all data underlying the findings in their manuscript fully available?

Reviewer #1: Yes

Reviewer #3: Yes

5. Is the manuscript presented in an intelligible fashion and written in standard English?

Reviewer #1: Yes

Reviewer #3: Yes

6. Review Comments to the Author

Reviewer #1: The author has adequately addressed the concerns of the reviewers. Actually, I had no issue as a reviewer for this paper. There is no ethical or other issue.

Reviewer #3: The revised version of the manuscript demonstrates significant improvements in both content and clarity. Considering the revisions made by the authors, the manuscript now meets the standards for publication. I appreciate the opportunity to participate in the review process and contribute to the evaluation of this manuscript.

7. PLOS authors have the option to publish the peer review history of their article (what does this mean?). If published, this will include your full peer review and any attached files.

Reviewer #1: **Yes: **Gour Gobinda Goswami

Reviewer #3: **Yes: **Md Lokman Hossain

---

## [Editor Report · Acceptance letter]

13 Mar 2024

PONE-D-23-30209R1 

PLOS ONE

Dear Dr. Hossain, 

I'm pleased to inform you that your manuscript has been deemed suitable for publication in PLOS ONE. Congratulations! Your manuscript is now being handed over to our production team.

Kind regards, 

on behalf of

Dr. Mohammed Sarfaraz Gani Adnan 

Academic Editor

PLOS ONE